# Boosting Mitochondrial Biogenesis Diminishes Foam Cell Formation in the Post-Stroke Brain

**DOI:** 10.3390/ijms242316632

**Published:** 2023-11-23

**Authors:** Sanna H. Loppi, Marco A. Tavera-Garcia, Natalie E. Scholpa, Boaz K. Maiyo, Danielle A. Becktel, Helena W. Morrison, Rick G. Schnellmann, Kristian P. Doyle

**Affiliations:** 1Department of Immunobiology, College of Medicine, University of Arizona, Tucson, AZ 85719, USA; sannaloppi@arizona.edu (S.H.L.); marcotavera@arizona.edu (M.A.T.-G.); maiyob1@arizona.edu (B.K.M.); dbecktel@arizona.edu (D.A.B.); 2Department of Pharmacology and Toxicology, College of Pharmacy, University of Arizona, Tucson, AZ 85719, USA; scholpa@pharmacy.arizona.edu (N.E.S.); schnell@pharmacy.arizona.edu (R.G.S.); 3College of Nursing, University of Arizona, Tucson, AZ 85719, USA; hmorriso@arizona.edu; 4BIO5 Institute, College of Medicine, University of Arizona, Tucson, AZ 85719, USA; 5R. Ken Coit Center for Longevity and Neurotherapeutics, College of Pharmacy, University of Arizona, Tucson, AZ 85719, USA; 6Department of Neurology, College of Medicine, University of Arizona, Tucson, AZ 85719, USA; 7Arizona Center on Aging, College of Medicine, University of Arizona, Tucson, AZ 85719, USA; 8Department of Psychology, College of Medicine, University of Arizona, Tucson, AZ 85719, USA; 9Department of Neurosurgery, College of Medicine, University of Arizona, Tucson, AZ 85719, USA

**Keywords:** stroke, aging, mitochondrial biogenesis, beta-2-adrenergic activation

## Abstract

Following ischemic stroke, the degradation of myelin and other cellular membranes surpasses the lipid-processing capabilities of resident microglia and infiltrating macrophages. This imbalance leads to foam cell formation in the infarct and areas of secondary neurodegeneration, instigating sustained inflammation and furthering neurological damage. Given that mitochondria are the primary sites of fatty acid metabolism, augmenting mitochondrial biogenesis (MB) may enhance lipid processing, curtailing foam cell formation and post-stroke chronic inflammation. Previous studies have shown that the pharmacological activation of the β2-adrenergic receptor (β2-AR) stimulates MB. Consequently, our study sought to discern the effects of intensified β2-AR signaling on MB, the processing of brain lipid debris, and neurological outcome using a mouse stroke model. To achieve this goal, aged mice were treated with formoterol, a long-acting β2-AR agonist, daily for two and eight weeks following stroke. Formoterol increased MB in the infarct region, modified fatty acid metabolism, and reduced foam cell formation. However, it did not reduce markers of post-stroke neurodegeneration or improve recovery. Although our findings indicate that enhancing MB in myeloid cells can aid in the processing of brain lipid debris after stroke, it is important to note that boosting MB alone may not be sufficient to significantly impact stroke recovery.

## 1. Introduction

Stroke is a leading cause of death and disability worldwide [1]. Following an ischemic stroke, the breakdown of myelin and other cell membranes exceeds the lipid processing abilities of both infiltrating macrophages and resident microglia. This results in the formation of foam cells within the infarct, which perpetuates a chronic inflammatory response and contributes to neurological damage (reviewed in [2]). 

In our prior research, we demonstrated that fatty acid β-oxidation, as evidenced by acyl carnitine levels, is notably elevated in myeloid cells in tissue affected by stroke for up to 4 weeks post-stroke [3]. This likely reflects the processing of myelin and other brain lipid debris by microglia and macrophages as they transition into foam cells.

Fatty acid oxidation primarily takes place in the mitochondria of a cell [4]. Therefore, increasing mitochondrial biogenesis (MB) could potentially help cells process fatty acids more efficiently and reduce foam cell accumulation and chronic inflammation after stroke.

Beta-2-adrenergic receptors (β2-ARs) are a type of G protein-coupled receptor found on the surface of various cell types including neurons, astrocytes, microglia, vascular cells, and immune cells [5,6,7,8]. They play a pivotal role in the sympathetic nervous system’s response by mediating the actions of catecholamines, specifically epinephrine and norepinephrine. When activated by these catecholamines, they set off a series of intracellular events leading to a range of physiological responses, from the relaxation of smooth muscles in the airways to vasodilation and increased heart rate [9,10]. Recent studies also indicate that β2-ARs have a role in mitochondrial biogenesis [11]. This is because β2-AR activation increases the activation of the PGC-1α (Peroxisome proliferator-activated receptor-gamma coactivator 1 alpha)/NRF1 (Nuclear respiratory factor 1)/TFAM (mitochondrial transcription factor A) pathway, a key transcriptional regulator of MB [12]. This pathway is known to regulate the expression of genes involved in mitochondrial DNA replication, transcription, translation, and assembly [13].

Pharmacological stimulation of the β2-AR has previously been shown to induce mitochondrial biogenesis and improve outcome in the injured central nervous system, but not in the context of stroke [14]. Therefore, to build upon these previous studies, we aimed to determine how increasing β2-adrenergic signaling after stroke impacts mitochondrial biogenesis, the processing of fatty acids derived from brain lipid debris, and the formation of foam cells in the brain in a mouse model of stroke. We hypothesized that increasing β2-adrenergic signaling after stroke would promote mitochondrial biogenesis, enhance fatty acid processing, and reduce foam cell formation, thereby improving neurological outcomes.

To test our hypothesis, we administered formoterol, a long-acting β2-AR agonist, daily for two and eight weeks after stroke in aged mice. We used aged mice to better represent the age of stroke patients who are more likely to experience poorer outcomes after stroke. As clinically relevant primary endpoints, we assessed infarct size and brain atrophy using magnetic resonance imaging (MRI), neurofilament light (NF-L) release in the plasma as a biomarker of neurodegeneration, and the overall effect on recovery using a horizontal ladder test and a nest building test. We also measured cyclic adenosine monophosphate (cAMP) levels to validate that our assessment of formoterol was at a dose capable of sustaining β2-adrenergic signaling for at least two weeks after stroke.

## 2. Results

### 2.1. Study Design

The timeline of the experiment is presented in Figure 1. Briefly, for Cohort 1, mice were treated with 0.3 or 1.0 mg/kg of formoterol or vehicle daily, with treatment beginning 24 h after stroke. Mice were sacrificed at 14 days and their brains were processed for Western Blot, immunostaining, boron-dipyrromethene (BODIPY) staining, and the measurement of cAMP and acyl carnitine levels. Their plasma was processed for NF-L quantitation. In Cohort 2, mice were administered 0.3 mg/kg of formoterol or vehicle daily starting 24 h post-stroke and were sacrificed at 8 weeks post-stroke. Behavioral assessments were conducted pre-stroke, 24 h post-stroke, and then weekly. At 3 days post-stroke, infarct size was determined using MRI. Ventricle size was evaluated by MRI as a marker of brain atrophy at both 2 and 8 weeks post-stroke.

### 2.2. Formoterol Administration Induces MB in the Infarct

Upon binding to the β2-AR, formoterol activates the Gα-subunit, which in turn stimulates adenylate cyclase [11]. This cascade leads to an accumulation of cyclic adenosine monophosphate (cAMP) as depicted in Figure 2A. Our metabolomic analysis revealed that when mice were administered formoterol at 0.3 mg/kg for two weeks after stroke, the concentration of cAMP in the infarct was increased compared to vehicle-treated mice (Figure 2B). However, this rise in the cAMP concentration was not observed at a 1.0 mg/kg dose. Notably, while the activation of the Gα subunit is not required for MB, the release of the Gβγ heterodimer post formoterol–β2AR binding plays a crucial role [11,15] (Figure 2A).

The release of the Gβγ heterodimer stimulates phosphoinositide 3-kinase (PI3K), which activates the serine/threonine kinase Akt [11]. This sequence of events leads to the phosphorylation of endothelial nitric oxide synthase (eNOS), triggering nitric oxide (NO) production. Consequently, NO activates soluble guanylyl cyclase (sGC), causing an accumulation of cyclic guanosine monophosphate (cGMP). This cascade results in increased expression of TFAM and increased levels of NADH-ubiquinone oxidoreductase 75 kDa subunit, mitochondrial (NDUSF1), a marker of MB [16,17] (Figure 2A).

Mice treated with 0.3 mg/kg of formoterol for 2 weeks exhibited significant increases in mitochondrial DNA (mtDNA) copy numbers and elevated levels of TFAM and NDUSF1 within the infarcted region (Figure 2C). Extending this treatment to 8 weeks saw a rise in mtDNA copy numbers and TFAM, with a noticeable trend towards higher NDUSF1 levels (Figure 2D). Yet, when the dosage was amplified to 1.0 mg/kg over a 2-week span, these enhancements were no longer statistically significant (Figure 2E). On the contralateral side, the expression levels of these proteins were not different between the vehicle- and formoterol-treated groups, with the exception that TFAM was significantly elevated in mice receiving 0.3 mg/kg formoterol for 2 weeks (Appendix A). These data demonstrate that treating mice with 0.3 mg/kg of formoterol induces mitochondrial biogenesis in the infarct, and that treatment with 1.0 mg/kg has a diminished effect, perhaps due to β2-AR receptor desensitization in response to excessive stimulation.

### 2.3. Formoterol Administration Alters Phagocytic Cell Morphology in the Infarcted Region

To evaluate the effect of 2 weeks of formoterol treatment on microglia and macrophage morphology, we used Iba-1 as a pan marker for microglia and macrophages [18], and CD68 to identify activated, phagocytotic microglia or macrophages [19]. In the area of gliosis adjacent to the infarct core (depicted in Figure 3A), there was an increased number of Iba-1-positive cells in formoterol-treated mice compared to the vehicle-treated group (Figure 3B,C). Furthermore, the process length of these cells was shorter in the formoterol group (Figure 3D). The immunoreactive area associated with CD68 expression was significantly larger in mice treated with formoterol than in those receiving the vehicle (Figure 3E,F). This suggests that the microglia and macrophages in the infarcted area of the brains of formoterol-treated mice are more activated, more phagocytotic, and more numerous than in vehicle-treated mice. Meanwhile, GFAP staining, indicative of activated astrocytes [20], revealed neither morphological nor expression level differences between the two groups (Figure 3G,H).

### 2.4. Lipid Droplets Were Significantly Reduced in the Infarcts of Mice Receiving Formoterol

BODIPY 492/515, a green, fluorescent lipophilic dye, is frequently employed to label lipid droplets [21]. Given our observations suggesting enhanced phagocytic activity in the microglia and macrophages of formoterol-treated mice, we aimed to assess lipid droplet accumulation within the infarcted regions of these mice. To account for potential autofluorescence in stroke-infarcted regions, we imaged unstained brain sections in parallel with BODIPY-stained sections, using the same fluorescence channel (Figure 4A). No fluorescence was observed in the unstained sections at an exposure time of 1/150 s (first image in Figure 4A). Detectable autofluorescence in the unstained section necessitated an extended exposure of 1.2 s (second image in Figure 4A). However, the BODIPY-stained sections displayed robust fluorescence even at the shorter 1/150 s exposure (third image in Figure 4A). We noted a pronounced decrease in BODIPY fluorescence in the infarcted regions of formoterol-treated mice compared to controls (Figure 4B, with representative images in Figure 4C). Figure 4D presents a composite image of the infarcted region, highlighting Iba-1, CD68, and BODIPY staining, underscoring that the majority of the staining is intracellular and within microglia or macrophages. The contralateral area remained unstained with BODIPY. Collectively, these findings indicate that the microglia/macrophages in the infarcts of formoterol-treated mice contain fewer lipid droplets. This supports the hypothesis that they process brain lipid debris more efficiently than their counterparts in vehicle-treated mice.

### 2.5. Formoterol Reduced Levels of Acyl Carnitines in the Infarcted Region

Following an injury to the CNS, such as a stroke, damaged or apoptotic cells—including their myelin debris—are taken up by phagocytosis into phagolysosomes. Within these compartments, enzymes like lysosomal acid lipase degrade the ingested myelin lipids, releasing free fatty acids and other lipid derivatives [22,23]. Once liberated, these fatty acids must cross the lysosomal membrane to reach the cytosol. Here, they bind to fatty acid-binding proteins (FABPs), which facilitate their transport to various cellular destinations: the mitochondria for β-oxidation, the endoplasmic reticulum for lipid synthesis, or other sites for lipid-mediated signaling [24,25].

For β-oxidation in the mitochondria, cytosolic fatty acids are first activated to fatty acyl-CoA molecules by the enzyme acyl-CoA synthetase (also known as fatty acid CoA ligase) [26,27]. Next, on the outer mitochondrial membrane, carnitine palmitoyltransferase I (CPT1) transfers the fatty acyl group from CoA to carnitine, producing acyl carnitine. This acyl-carnitine is shuttled across the inner mitochondrial membrane by a translocase protein, which, in the process, exports a free carnitine molecule from the mitochondria to the intermembrane space. Once within the mitochondrial matrix, carnitine palmitoyltransferase II (CPT2) reconverts acyl carnitine back to fatty acyl-CoA and a free carnitine molecule. The fatty acyl-CoA then undergoes β-oxidation, sequentially cleaving two-carbon units in the form of acetyl-CoA, while also producing NADH and FADH2. These acetyl-CoA units can subsequently enter the citric acid (Krebs) cycle, driving further oxidative metabolism and energy generation [28,29] (Figure 5A).

Considering this pathway, we assessed acyl carnitine and carnitine levels as markers for fatty acid beta-oxidation and the metabolic breakdown of myelin and other brain lipid debris. As anticipated, and as we have shown previously [3], acyl carnitine levels in the infarcted region of the ipsilateral hemisphere of the vehicle-treated mice were markedly elevated compared to those in the equivalent region of the contralateral hemisphere. Contrastingly, formoterol-treated brains exhibited a significant reduction in acyl carnitine levels in the infarcted region compared to those treated with the vehicle (Figure 5B). Carnitine concentrations followed a similar pattern; they were notably augmented in the stroked tissue, but formoterol treatment aligned these levels more closely with those in the unaffected, contralateral cortex (Figure 5C).

Cumulatively, these results support that two weeks of formoterol treatment post-stroke optimizes fatty acid metabolism and enhances the clearance of brain lipid debris within the infarcted region.

### 2.6. Formoterol Did Not Alter Infarct Size, Ventricle Size, or Plasma NF-L Release

Having observed changes in lipid processing after two weeks of treatment with 0.3 mg/kg of formoterol, we sought to evaluate its impact on post-stroke neurodegeneration. Neurofilament light chain (NF-L) is one of the subunits of neurofilaments, which are intermediate filament proteins found in neurons. Elevated levels of NF-L in blood plasma are associated with neuronal damage or axonal injury [30]. Accordingly, we measured NF-L concentrations in the plasma of the mice that underwent vehicle or formoterol treatment at 2 weeks post stroke. While stroke markedly raised NF-L plasma levels in comparison to age-matched naive mice, formoterol administration yielded no significant changes (Figure 6A). We then assessed infarct size at 3 days following stroke, and ventricle size at 2 weeks and 8 weeks after stroke as a marker of brain atrophy. No discernable differences were detected between the groups for any evaluated timepoint (Figure 6B–D). Together, these results indicate that formoterol did not have a notable effect on post-stroke neurodegeneration.

### 2.7. Formoterol Treatment Did Not Alter Motor Recovery

We assessed the motor recovery of the mice using the ladder and nest building tests. In the ladder test, stroked mice performed notably worse than age-matched naïve controls. However, formoterol treatment did not enhance recovery compared to vehicle-treated mice, as determined by repeated measures ANOVA (Figure 7A). Similarly, in the nest building test, both the vehicle- and formoterol-treated mice exhibited significant deficits on day 1 post-stroke compared to age-matched naïve mice. Yet, no discernible difference between the formoterol and vehicle groups was identified using repeated measures ANOVA (Figure 7B).

## 3. Discussion

The effect of β2-AR activation on stroke outcomes is complex and has yielded mixed results in the literature. This may be due to a number of factors, including the different stroke models used, the age of the animals, the method of activation or inactivation, and the starting time of the treatment in relation to the time of the ischemia.

Some studies have shown that β2-AR activation can be beneficial in stroke. For example, Culmsee et al. (1999) demonstrated that pretreatment with a β2-AR agonist could decrease infarct size in both mice and rats [31]. However, other studies have shown that β2-AR activation can be harmful in stroke. For example, Lechtenberg et al. (2019) observed that treatment with clenbuterol 3 h post-photothrombotic stroke not only enlarged the lesion size but also altered microglial morphology and reduced TNFα and IL-10 expression in mice [32]. Similarly, Sun et al. (2017) discovered that β2-AR blockade can mitigate blood–brain barrier (BBB) damage during acute cerebral ischemia in mice, suggesting that β2-AR activation can contribute to BBB damage [33]. 

However, our research uniquely explores the chronic effects of β2-AR activation post-stroke on MB, lipid metabolism, and foam cell formation. Previous works from our team and others have demonstrated formoterol’s capacity to induce MB across diverse cell types, encompassing renal proximal tubule cells, cardiomyocytes, and skeletal muscle cells, as well as in various disease models [34]. For instance, formoterol treatment enhanced mitochondrial function and ATP production in a mouse model of acute kidney injury [35]. Furthermore, a 2019 publication from our group highlighted the benefits of formoterol in reducing neuronal death, augmenting motor function, and promoting MB in a mouse model of SCI, while also attenuating spinal cord inflammation [14]. Moreover, studies like the one by Lee et al. (2013) have identified formoterol’s potential to augment fatty acid oxidation in human tissue [36].

In this study, we administered formoterol to aged mice after experimental stroke and observed that the 0.3 mg/kg dose yielded positive results, including increased mitochondrial biogenesis and reduced foam cell accumulation. Interestingly, the higher dose of 1.0 mg/kg did not replicate these effects, which raises the possibility of β2-AR desensitization at elevated doses [37,38]. 

Following two weeks of treatment, a decrease in acyl-carnitine levels within the infarcted brain region was evident, suggesting a modulation in lipid metabolism. While this could mean that formoterol accelerates lipid metabolism in microglia and macrophages, another interpretation revolves around the inherent metabolic dysfunction within foam cells [39]. This dysfunction could lead to the incomplete beta-oxidation of fatty acids and result in acyl-carnitine accumulation in the infarcted region. Supporting this possibility, elevated acyl-carnitine levels have been associated with an incomplete fatty acid oxidation process [40]. In such a scenario, the intervention of formoterol might serve to re-establish mitochondrial functionality in microglia and macrophages, thereby facilitating the complete beta-oxidation of fatty acids.

Despite these potential therapeutic mechanisms, our MRI and NF-L assay analyses showed no neuroprotective effects of formoterol. Furthermore, there was no observable difference in motor recovery post-stroke between formoterol-treated mice and control mice. Given the known side effects of formoterol, such as tachycardia and tremors, and its association with worsened cardiac function, the adverse effects might overshadow its benefits, especially in aged mice [41]. 

While our previous publication demonstrated the effectiveness of formoterol in reducing neuronal death and enhancing motor function in SCI, its lack of similar efficacy in stroke could be due to the differences between these conditions. For example, the distinct cellular compositions and pathophysiological processes of the brain and spinal cord, along with variations in inflammatory and immune responses, differing regenerative capabilities, and unique secondary complications in stroke compared to SCI could all play a role in this observed discrepancy. These differences emphasize the importance of testing therapies specifically for each condition rather than extrapolating efficacy from one to the other. Additionally, it is important to consider the variation in the experimental models used; in the SCI study, the subjects were 8–9-week-old female mice, while in this stroke study, we used 18–20-month-old male mice.

It is also important to note that formoterol is a potent β2-AR agonist, and research by McCulloch et al. has revealed that adrenergic signaling, triggered by the surge in sympathetic activity after stroke, decreases the number of B cells in the periphery and contributes to post-stroke immunosuppression [42]. This immunosuppressive effect may be amplified by formoterol administration after stroke. 

In conclusion, our study sheds light on the potential of mitochondrial function enhancement in microglia and macrophages as a therapeutic strategy post-stroke. While formoterol did not improve structural or motor recovery, it could be valuable in a multi-drug therapy. Further research is required to determine the optimal dose, timing, and safety of formoterol treatment after stroke, as well as to explore alternative methods of enhancing mitochondrial function in myeloid cells.

## 4. Materials and Methods

### 4.1. Animals

In total, 86 aged (18- to 20-month-old) male C57BL/6J mice, sourced from the aged rodent colony at the National Institute on Aging, were used in this study. The mice were housed in a temperature-controlled facility with 12 h light/dark cycles and free access to food and water. They were divided randomly (using GraphPad Prism Quick Calcs, Boston, MA, USA; https://www.graphpad.com/quickcalcs/randomize1/, accessed on 26 March 2021) into groups receiving either Formoterol (0.3 mg/kg or 1 mg/kg) or vehicle (DMSO) for 14 days following stroke, or into groups receiving Formoterol (0.3 mg/kg) or vehicle (DMSO) for 8 weeks after stroke. Six of these mice were harvested as naïve controls without any surgeries or treatments. The pre-set exclusion criteria of the study were (1) unsuccessful induction of ischemia (5 mice), (2) death of the animal during the experiment (8 mice), and (3) being a statistically significant outlier in any of the analyses (4 mice). All animal experiments followed the NIH guidelines and were approved by the University of Arizona Institutional Animal Care and Use Committee.

### 4.2. Stroke Surgeries

Permanent distal middle cerebral artery occlusion (dMCAO) was performed following previously established methods [43]. Mice were anesthetized using 3% isoflurane (JD Medical, Phoenix, AZ, USA), which was maintained during surgery. Upon exposing and retracting the temporalis muscle, a 1 mm hole was drilled into the temporal bone to reveal the MCA. After dura removal, the artery was cauterized. The temporalis muscle was then repositioned, and the incision sealed with surgical glue. Throughout the operation, a heating pad with a rectal probe ensured the animals’ body temperature remained at 37 ± 0.5 °C. Both respiration and temperature were consistently monitored. Immediately after surgery, mice were placed in a hypoxia chamber (Coy Laboratory products, Grass Lake, MI, USA) containing 9% oxygen and 91% nitrogen for 45 min. The purpose of hypoxia in this model is to both increase infarct size and reduce variability in infarct size [43]. A single dose of buprenorphine hydrochloride (Buprenex^®^ Injection 0.3 mg/mL, Henry Schein Medical, Melville, NY, USA; 0.1 mg/kg s.c.) was administered prior to surgery, and sustained release buprenorphine (Buprenorphine SR 1 mg/mL, Zoopharm LLC, Laramie, WY, USA; 1 mg/kg s.c.) was administered 24 h after surgery as a post-operative analgesic. For pain management, buprenorphine hydrochloride (Buprenex^®^ Injection 0.3 mg/mL, Henry Schein Medical, Melville, NY, USA; 0.1 mg/kg s.c.) was given before the procedure. A subsequent dose of sustained-release buprenorphine (Buprenorphine SR 1 mg/mL, Zoopharm LLC, Laramie, WY, USA; 1 mg/kg s.c.) was provided 24 h post-surgery.

### 4.3. Perfusion and Tissue Collection

Fourteen days or eight weeks post-stroke, mice were anesthetized using 3% isoflurane. Blood samples were then drawn from the heart, followed by a transcardial perfusion using a 0.9% saline solution. For immunohistochemistry (IHC), whole brains were extracted and immediately fixed in 4% PFA in PBS for 22 h at 4 °C. Subsequently, they were immersed in a 30% sucrose solution at 4 °C for a minimum of 48 h before further processing. Using a Microm HM 450 sliding microtome (Thermo Fisher Scientific, Waltham, MA, USA), coronal sections with a thickness of 40 µm were prepared spanning the entire brain. For biochemical evaluations, the infarct, peri-ischemic regions, and their equivalent areas on the contralateral side were excised and rapidly frozen in liquid nitrogen. Blood samples from control mice were collected in a similar fashion. Once drawn, the blood was centrifuged at 10,000 rpm for 15 min at 4 °C. The plasma layer was carefully collected, ensuring no disturbance to other layers, and immediately frozen in liquid nitrogen.

### 4.4. Immunostaining

Immunostaining was performed on free-floating or pre-mounted brain sections using standard protocols. Primary antibodies against CD68 (1:1000; Bio-Rad, Hercules, CA, USA, cat. MCA1957GA), ionized calcium-binding adapter molecule 1 (IBA-1, 1:500; Wako, Richmond, VA, USA, cat. 019-19741), and glial fibrillary acidic protein (GFAP; 1:2000; Millipore Sigma, Burlington, MA, USA, cat. AB5541) were used in conjunction with the appropriate secondary antibodies. Sections were imaged using a Leica DM6000B (Buffalo Grove, IL, USA) light microscope coupled with a Leica DFC 7000 T camera, or confocal microscope (Zeiss NLO 880, San Diego, CA, USA).

### 4.5. Microglia Morphology Analysis

For microglia morphology analysis, images were acquired on a confocal microscope (Zeiss NLO 880, San Diego, CA, USA) using a 40X objective (236.16 × 236.16-micron area) from the area of gliosis proximal to the core of the infarct and corresponding area of the contralateral side. Microglia morphology was analyzed from 8-bit images, as described previously [44]. Microglia morphology parameters (process length and number of endpoints) were summed, and all data were divided by the cell soma count per image frame to calculate the summed microglia process length/cell. The average values from each region within each animal were used for statistical analysis.

### 4.6. Boron-Dipyrromethene (BODIPY) 492/515 Staining Procedure 

Tissue sections were allowed to air-dry overnight at room temperature (RT). Following the drying, sections underwent a series of washing steps in fresh 0.1 M PBS at RT. Each wash lasted 10 min, during which gentle agitation was applied, and any excess PBS was carefully removed after each cycle. For staining, a working solution of BODIPY 492/515 was formulated by diluting the stock (0.1 mg/mL in dimethyl sulfoxide, DMSO) to a final concentration of 1:25. The tissue sections were then submerged in this solution and incubated for 30 min in a dark environment at RT. Post-incubation, coverslips were applied using Vectashield HardSet Antifade mounting media (Vector Laboratories, Newark, CA, USA) to prepare the samples for subsequent fluorescence microscopy analysis.

### 4.7. Magnetic Resonance Imaging (MRI)

Infarct volumes and ventricle sizes were assessed by magnetic resonance imaging (MRI) at time points 3 days, 2 weeks, and 8 weeks after stroke using a Bruker Biospec 70/20 7.0 T scanner with ParaVision-360.2.0 software and a 4-channel phase array mouse coil (Bruker Biospin GmbH, Ettlingen, Germany). Mice were placed in a cradle equipped with a stereotaxic frame, an integrated heating system to maintain body temperature at 37 ± 1 °C, and a pressure probe to monitor respiration. During MRI acquisition, anesthesia was maintained via the inhalation of 1.5–3% isoflurane. High-resolution structural images were acquired using a T2-weighted RARE multi-echo Bruker pulse sequence with the following parameters: repetition time (TR) = 2000 ms; flip angle = 180°; RARE factor = 12; matrix size = 128 × 128; averages = 2; field of view (FOV) = 1.92 cm × 1.92 cm; slice thickness = 0.8 mm; number of slices = 15; acquisition time = 8 min 53 s. Infarcts and hemispheric cross sections were manually delineated on T2-weighted images using Mango v4.1. Infarct volumes were calculated using the following equation: overall infarct volume × (1 − [ipsilateral hemisphere volume − contralateral hemisphere volume]/contralateral hemisphere volume) [45], after which they were normalized to vehicle. Ventricle sizes were measured with Mango and subsequently normalized to the contralateral ventricles. Twelve slices were analyzed per mouse for each parameter.

### 4.8. Western Blot and mtDNA

For Western Blot, protein was extracted from the stroke infarcts and corresponding contralateral tissue using RIPA buffer with protease inhibitor cocktail (1:100), 1 mM sodium fluoride, and 1 mM sodium orthovanadate (Sigma-Aldrich, St. Louis, MO, USA), as described previously [14]. Protein was quantified using a bicinchoninic acid assay, and 10 μg of protein was separated via electrophoresis using 4–15% SDS-PAGE gels, and then transferred to nitrocellulose membranes (Bio-Rad, Hercules, CA, USA). Membranes were blocked in 5% milk in TBST and incubated overnight with primary antibodies with constant agitation at 4 °C. Membranes were incubated with the appropriate horseradish peroxidase-conjugated secondary antibody and visualized using chemiluminescence (Thermo Scientific, Waltham, MA, USA) on a GE ImageQuant LAS4000 (GE Life Sciences, Pittsburg, PA, USA). Optical density was determined using Image Studio Lite software (v5.0, LI-COR Biosciences, Lincoln, NE, USA). Primary antibodies used were TFAM, NDUFS1 (1:1000), and α-tubulin (1:10,000, Abcam, Cambridge, UK).

To assess mitochondrial DNA (mtDNA) expression, we isolated DNA from the stroke infarcts and their corresponding contralateral tissue using a Qiagen DNeasy Blood and Tissue Kit (Valencia, CA, USA). Subsequently, we utilized 5 ng of the extracted DNA for quantitative PCR (qPCR) to measure the relative mtDNA content as previously described [14]. Mitochondrial gene ND1 was measured and normalized to the nuclear encoded gene beta-actin. The primers used were as follows: ND1 sense 5′-TGAATCCGAGCATCCTACC-3′, ND1 antisense 5′-ATTCCTGCTAGGAAAATTGG-3′, beta-actin sense 5′-GGGATGTTTGCTCCAACCAA-3′, and beta-actin antisense 5′GCGCTTTTGACTCAGGATTTAA-3′.

### 4.9. Ladder Test

Motor recovery following stroke was evaluated using the Ladder test [46] on days 1 and 7 post-stroke and then weekly up to 8 weeks post-stroke. The Ladder test apparatus had plexiglass rungs, each 10.16 cm long and 3.2 mm in diameter, spaced 1.6 cm apart. These rungs were situated between two plexiglass walls, each 76.20 cm in length and 15.24 cm in height. The walls were set 3.18 cm apart, allowing ample room for the mice to traverse. The entire setup was elevated 18 cm above the table using blocks. Tests were conducted in a dimly lit room. A desk lamp lit the starting zone to motivate the mice to cross the ladder towards a small box at the end zone. At the commencement of the test, mice were positioned in a plastic chamber aligned with the ladder’s starting rungs. As each mouse navigated the ladder, its footfalls were captured by a camera, positioned beneath, and moved synchronously with the mouse along the table. Before official testing, each mouse underwent 8 training trials to attain a performance error rate of ≤10–12%. On actual test days, each mouse was evaluated twice, and the resulting scores were averaged. Between each trial, the ladder was cleaned to eliminate olfactory cues. Video footage of each mouse’s traversal was reviewed at playback speeds of 0.25−0.30× using standard video playback software (VLC v3.0.18, VideoLAN, Paris, France). A skilled observer examined the front limb contralateral to the stroke infarct to tally correct and incorrect steps. Videos were scored independently by two trained examiners, with their scores averaged for final analysis. The percentage of accurate foot placements was determined as 100 × (number of correct steps/(number of correct steps + number of missteps)).

### 4.10. Nest Building Test

The nest building test was administered to evaluate the mice at 1 day and 1 week post-ischemia, followed by weekly assessments up to the 8-week mark. Each mouse was individually housed in a cage containing a cotton nestlet. They were permitted to engage with the nestlet overnight. The following morning, the degree of nestlet shredding was assessed. Scoring ranged from 0 (nestlet untouched) to 5 (nestlet fully shredded and nest constructed), as detailed in Deacon (2006) [47].

### 4.11. Neurofilament Light Assay

Neurodegeneration levels were evaluated using terminal plasma samples at 2 weeks following stroke. These samples were forwarded to PBL Assay Science (Piscataway, NJ, USA) for analysis using the Simoa™ NF-Light^®^ kit (Cat #103186, Quanterix, Billerica, MA, USA). Plasma from age-matched naïve mice (aligned in age with the formoterol and vehicle-treated groups) served as a control.

### 4.12. Measurement of cAMP and Acyl Carnitines

An analysis of cAMP and acyl carnitine levels in brain tissue was performed by Metabolon Inc. (Morrisville, NC, USA) as previously described [3]. Briefly, brain tissue underwent cAMP and acyl carnitine profiling by Metabolon Inc. using UPLC-MS/MS. Upon receipt, samples were stored at −80 °C and prepared using the MicroLab STAR^®^ system. Proteins were precipitated with methanol to recover diverse metabolites, followed by fractionation for different analysis modes. Organic solvents were evaporated using TurboVap^®^, and sample extracts were stored under nitrogen before analysis.

Multiple controls were employed: a pooled sample for technical replication, water samples as process blanks, and QC standards spiked into every sample for instrument performance and chromatographic alignment. Instrument and process variabilities were determined via median RSD calculations. The UPLC-MS/MS utilized Waters ACQUITY UPLC and Thermo Scientific Q-Exactive spectrometer with a HESI-II source.

Data extraction involved Metabolon’s systems to peak-identify and QC process. cAMP and the acyl carnitines were identified against Metabolon’s library of authenticated standards, considering retention time/index, mass-to-charge ratio, and MS/MS spectral data.

### 4.13. Statistics

Statistical analyses were performed using GraphPad Prism software v9.3.1 (GraphPad Software, LaJolla, CA, USA), and normality was assessed using the Kolmogorov–Smirnov test. Statistical tests and sample sizes are provided in each figure legend, and *p* values less than 0.05 were considered to be statistically significant. Statistically significant outliers, calculated using GraphPad Prism QuickCalcs (Boston, MA, USA; https://www.graphpad.com/quickcalcs/Grubbs1.cfm, accessed on 13 June 2022), were excluded from the datasets. Data are presented as mean ± SD. Median-scaled raw data were used to generate the heat maps in Figure 5 and Welch’s two-sample *t*-test was used to test whether two unknown means were different between two independent populations.

## Figures and Tables

**Figure 1 ijms-24-16632-f001:**
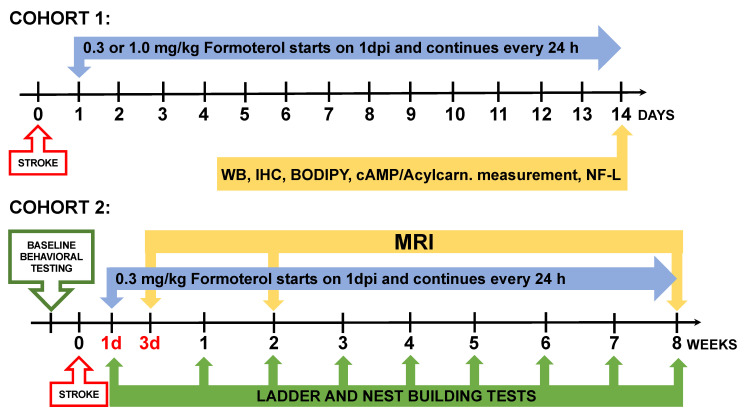
Study design. Cohort 1 received vehicle, 0.3 mg/kg formoterol, or 1.0 mg/kg formoterol for 2 weeks after stroke. Based on the results of Cohort 1, the 0.3 mg/kg dose was selected for Cohort 2, who received formoterol for 8 weeks. The 8-week study was designed to evaluate the long-term effects of formoterol treatment on stroke recovery. Outcomes included infarct size and brain atrophy assessment by MRI, and behavioral testing. WB = Western Blot, IHC = Immunohistochemistry, cAMP = cyclic adenosine monophosphate, Acylcarn = Acyl carnitines, NF-L = Neurofilament Light Assay.

**Figure 2 ijms-24-16632-f002:**
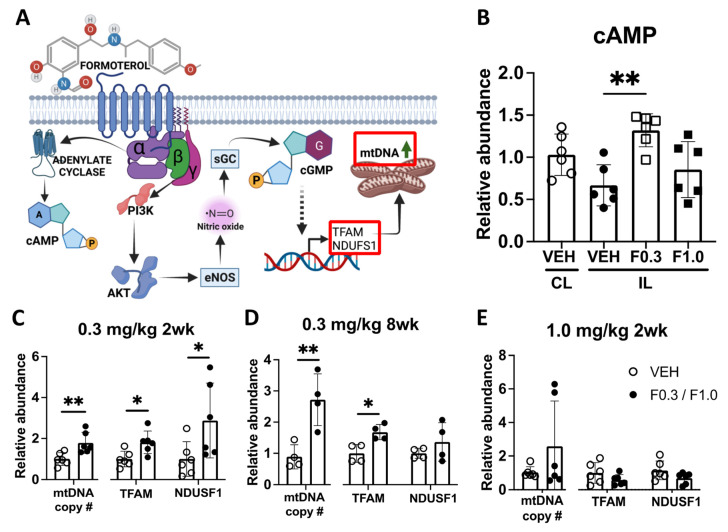
Formoterol administered at 0.3 mg/kg, but not at 1.0 mg/kg, increased cAMP levels and MB in the infarcted area. (**A**) A schematic illustration of the proposed mechanism by which formoterol induces MB (according to [11]). cAMP = cyclic adenosinemonophosphate, PI3K = phosphoinositide 3-kinase, AKT = serine/threonine kinase, eNOS = endothelial nitric oxide synthase, sGC = soluble guanylyl cyclase, cGMP = cyclic guanosine monophosphate, TFAM = mitochondrial transcription factor A, NDUSF1 = mitochondrial NADH-ubiquinone oxidoreductase 75 kDa subunit, mtDNA = mitochondrial DNA (created with BioRender.com). (**B**) Formoterol administered at 0.3 mg/kg, but not 1.0 mg/kg, elevated cAMP levels in the infarcted area of the brain; F*(3, 17.31) = 6.887 Brown–Forsythe ANOVA test followed by Dunnett’s T3 multiple comparison test, ** adj.*p* = 0.0026, *n* = 6, VEH = Vehicle, F0.3 = Formoterol 0.3 mg/kg, F1.0 = Formoterol 1.0 mg/kg, CL = contralateral, IL = ipsilateral. This indicates that 0.3 mg/kg formoterol is able to activate β2-AR signaling in the brain. (**C**) Formoterol 0.3 mg/kg elevated mtDNA copy # (mitochondrial DNA copy number) (t(10) = 3.190, ** *p* = 0.0097 unpaired two-tailed *t*-test), TFAM (t(10) = 2.967, * *p* = 0.0141 unpaired two-tailed *t*-test) and NDUSF1 (t(10) = 2.283, * *p* = 0.0456 unpaired two-tailed *t*-test, *n* = 6) after 2 weeks of treatment, and this effect was still visible with mtDNA copy number (t(6) = 3.993, ** *p* = 0.0072 unpaired two-tailed *t*-test) and TFAM (* *p* = 0.0286 two-tailed Mann–Whitney test, *n* = 4) at 8 weeks (**D**). The 1.0 mg/kg dose of formoterol failed to induce these markers of MB (**E**). Data presented as mean ± SD.

**Figure 3 ijms-24-16632-f003:**
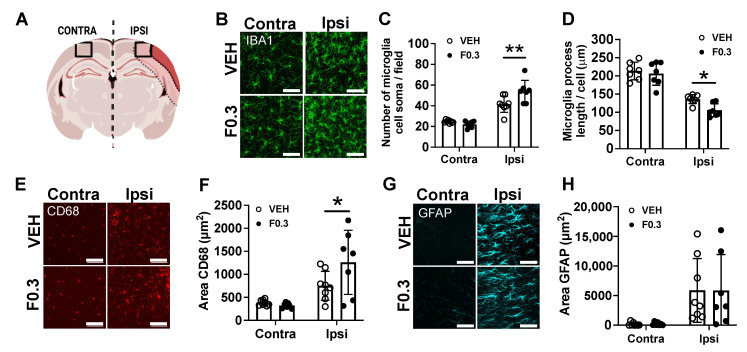
Formoterol 0.3 mg/kg for 2 weeks increased microglia but not astrocyte responses to ischemic stroke. (**A**) Microglia and astrocytes were imaged in the indicated regions after ischemic stroke (created with BioRender.com). (**B**) Representative images of Iba-1-positive microglia and corresponding summary data of increased microglia cell soma counts per frame (**C**) in formoterol-treated mice post-stroke; F(1, 26) = 9.174 *p* = 0.0055 interaction effect, F(1, 26) = 96.08 *p* < 0.0001 stroke effect, F(1, 26) = 3.924 *p* = 0.0583 treatment effect of two-way ANOVA followed by Sidak’s post hoc, ** adj.*p* = 0.0030. (**D**) Morphology analysis of Iba-1-positive cells. Mice treated with formoterol showed decreased microglia process length/cell in the infarcted area; F(1, 24) = 105.9 *p* < 0.0001 stroke effect, F(1, 24) = 4.737 *p* = 0.0396 treatment effect of two-way ANOVA followed by Sidak’s post hoc, * adj.*p* = 0.0388. (**E**) Representative images of CD68 and corresponding summary data of increased area of positive immunoreactivity in formoterol-treated mice post-stroke (**F**); F(1, 26) = 4.327 *p* = 0.0475 interaction effect, F(1, 26) = 22.65 *p* < 0.0001 stroke effect of two-way ANOVA followed by Sidak’s post hoc, * adj.*p* = 0.0284. (**G**) Representative images of GFAP and corresponding summary data of GFAP immunoreactivity area, increased post-stroke but unchanged with formoterol treatment (**H**). *n* = 7–8 per group. Contra = contralateral side, Ipsi = ipsilateral side, VEH = vehicle, F0.3 = Formoterol 0.3 mg/kg for 2 weeks. Scale bar = 50 µm. Data presented as mean ± SD.

**Figure 4 ijms-24-16632-f004:**
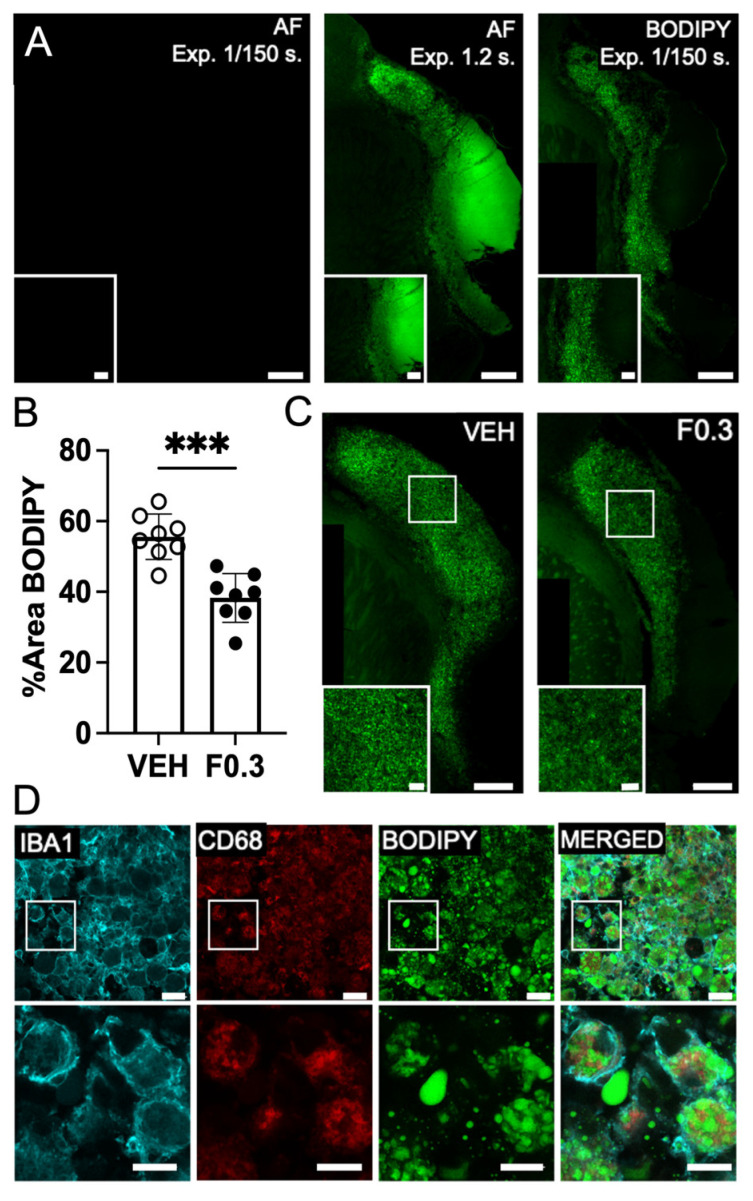
BODIPY staining was reduced in the infarcts of mice treated with formoterol 0.3 mg/kg for 2 weeks after stroke. (**A**) Representative images showing the difference between autofluorescence and BODIPY staining. Images were captured at 20× magnification, with inserts highlighting the specificity of the staining. The first autofluorescence (AF) image and the BODIPY were taken with an exposure time (Exp.) of 1/150 s, while the second AF image was taken with an exposure time of 1.2 s. Scale bars: 100 µm for the entire infarct and 30 µm for the inserts. (**B**) Quantification of BODIPY staining in stroke infarcts at 2 weeks post stroke. BODIPY staining was significantly reduced in the formoterol-treated group; t(14) = 5.19, *** *p* = 0.0001 unpaired two-tailed *t*-test, *n* = 8. Data presented as mean ± SD. (**C**) Representative immunofluorescent images of BODIPY staining in the infarct. Images at 2 weeks post stroke were captured at 20× magnification, with inserts highlighting the infarct region. Scale bars: 100 µm for the entire infarct and 20 µm for the inserts. VEH = vehicle, F0.3 = formoterol 0.3 mg/kg for 2 weeks. (**D**) Immunofluorescent staining demonstrates individual as well as colocalized staining of IBA1 (cyan), CD68 (red), and BODIPY (green) in the infarct. Images were captured at 40× magnification (zoomed 2×), showcasing cellular colocalization within the infarct. Scale bars: 20 µm for the main images and 10 µm for the inserts.

**Figure 5 ijms-24-16632-f005:**
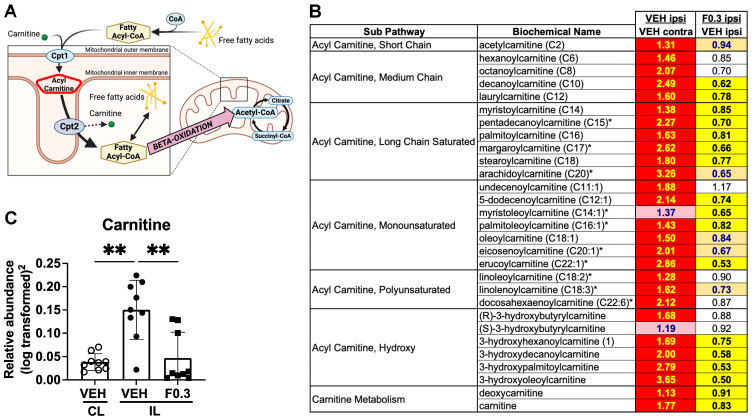
Formoterol treatment reduced levels of acyl carnitines in the stroked brain. (**A**) A schematic illustration of transportation of fatty acids into mitochondria for beta-oxidation (created with BioRender.com). (**B**) A heatmap showing that the levels of almost all of the detected acyl carnitines were elevated in the ipsilateral (IL) side of the vehicle-treated brains when compared to the contralateral side. However, the acyl carnitines in the ipsilateral side of the formoterol-treated brains were markedly reduced when compared to the ipsilateral side of brains receiving vehicle. Red-shaded cells indicate a significant increase (*p* < 0.05) in the metabolite in that comparison, and yellow-shaded cells indicate a significant decrease. Light red- and light orange-shaded cells indicate a trend (0.05 < *p* < 0.1) towards the respective direction, and the values represent fold change. (**C**) Carnitine levels increased after stroke, similar to the acyl carnitines, and formoterol treatment reduced them to contralateral side levels. CL = contralateral, IL = ipsilateral. F*(2, 17.12) = 14.14 *p* = 0.0002 Brown–Forsythe ANOVA test followed by Dunnett’s multiple comparisons test, ** adj.*p*(VEH CL vs. VEH IL) = 0.0019; ** adj.*p*(VEH IL vs. F0.3 IL) = 0.0059, *n* = 9, data presented as mean ± SD.

**Figure 6 ijms-24-16632-f006:**
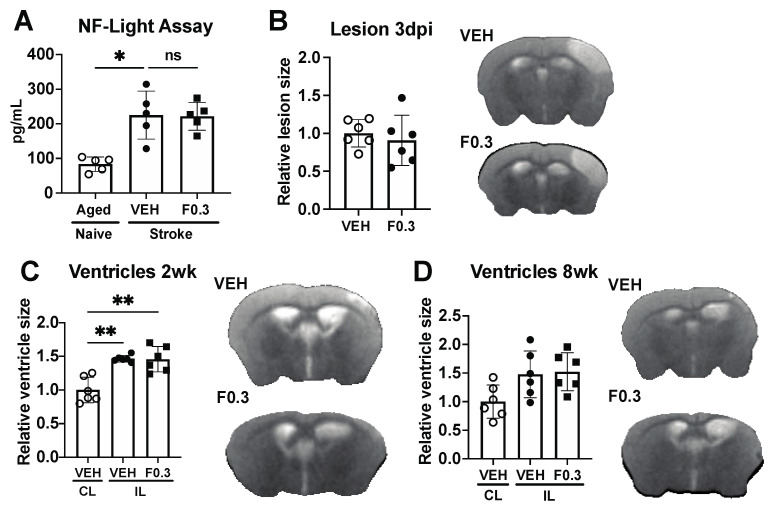
Formoterol did not affect neurodegeneration. (**A**) NF-L levels in plasma from vehicle (VEH)- and formoterol 0.3 mg/kg (F0.3)-treated mice were significantly elevated after stroke compared to age-matched naïve mice. Formoterol treatment did not further affect NF-L levels; F*(2, 7.266) = 14.21 Brown–Forsythe ANOVA followed by Dunnett’s T3 multiple comparisons test, * adj.*p* = 0.0192, ns = not significant, *n* = 5. (**B**) There was no difference in the size of the lesion at 3 days after ischemia. (**C**) Ventricle size at 2 weeks after stroke was significantly larger in the ipsilateral (IL) than the contralateral (CL) side, but there was no difference between the treatment groups; F*(2, 10.65) = 17.21 Brown–Forsythe ANOVA followed by Dunnett’s T3 multiple comparisons test, ** adj.*p*(VEH CL vs. VEH IL) = 0.0031, ** adj.*p*(VEH CL vs. F0.3 IL) = 0.0054, *n* = 6. (**D**) Ventricle sizes did not differ between the groups at the 8-week timepoint.

**Figure 7 ijms-24-16632-f007:**
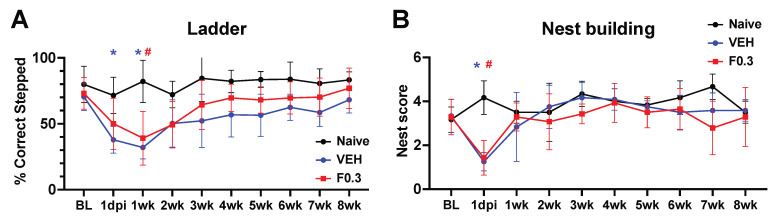
Formoterol did not affect motor recovery. (**A**) Both formoterol- and vehicle-treated mice showed similar stroke deficits during week 1 of the ladder test when compared to their age-matched naive counterparts; Blue * = VEH = vehicle, Red # = F0.3 = formoterol, F(18, 144) = 1.622 *p* = 0.0616 time x treatment, F(3.201, 51.21) = 10.05 *p* < 0.0001 time, F(2, 16) = 11.81 *p* = 0.0007 treatment, F(16, 144) = 5.477 *p* < 0.0001 subject effect of two-way repeated measures ANOVA followed by Tukey’s multiple comparisons test, 1dpi: blue * adj.*p* = 0.0190, 1 wk: blue * adj.*p* = 0.0102, red # adj.*p* = 0.0124. (**B**) The same applied to the nest building test. There was a significant stroke effect on day 1 after stroke, but the mice quickly recovered to the level of age-matched naïve animals; F(18, 117) = 2.518 *p* = 0.0016 time x treatment, F(3.697, 48.06) = 6.873 *p* = 0.0003 time, F(13, 117) = 5.374 *p* < 0.0001 subject effect of two-way repeated measures ANOVA followed by Tukey’s multiple comparisons test, blue * adj.*p* = 0.0246, red # adj.*p* = 0.0151. BL = Baseline measurement, 1 dpi = 1 day post ischemia, *n* = 4–8. Data presented as mean ± SD.

## Data Availability

The data used in this research are available upon request from the corresponding author, in compliance with the ethical and legal considerations governing data sharing and confidentiality.

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
