# Peer review of "Boosting Mitochondrial Biogenesis Diminishes Foam Cell Formation in the Post-Stroke Brain"

_ijms, 2023, doi:10.3390/ijms242316632_

Round 1

Reviewer 1 Report

Comments and Suggestions for Authors

The manuscript by Loppi et al explores the effects of formoterol treatment on mitochondrial biogenesis, the processing of brain lipid debris, and neurological outcomes using a mouse stroke model. In general, the manuscript is well written and easy to follow.  Although the treatment was not found to improve neurological outcomes or reduce infarct size, some changes were observed regarding mitochondrial beta oxidation, which may be of general interest and should be published. Some suggested improvements to the manuscript are provided below:

1)            Methods relating to the quantification of mitochondrial DNA copynumber are missing.

2)            It is unclear why data from contralateral hemisphere were not reported for mtDNA copynumber (Fig2), TFAM and NDUSF1 protein quantification (Fig2), or BODIPY (Fig3).

3)            In Fig2b, throughout Fig3, in Fig5c, and in Fig6a it is recommended that infarcted be changed to ipsi or ipsilateral for consistency. Also, in some cases IL/CL is used while in other Ispi/Contra is used. This should be made uniform throughout the manuscript and figures.

4)            The original blots that were provided are not labelled and are not referenced in the text. It is recommended that these blots be included in the supplement and referenced in the text.

Author Response

Response to comments on article “Boosting Mitochondrial Biogenesis Diminishes Foam Cell Formation in the Post-Stroke Brain” IJMS 2710076

Reviewer 1: The manuscript by Loppi et al explores the effects of formoterol treatment on mitochondrial biogenesis, the processing of brain lipid debris, and neurological outcomes using a mouse stroke model. In general, the manuscript is well written and easy to follow.  Although the treatment was not found to improve neurological outcomes or reduce infarct size, some changes were observed regarding mitochondrial beta oxidation, which may be of general interest and should be published. Some suggested improvements to the manuscript are provided below:

  • Methods relating to the quantification of mitochondrial DNA copy number are missing.

The authors thank the reviewer for noticing this omission. The method of quantifying the mtDNA copy number is now added to the Material and Methods on page 12.

2) It is unclear why data from contralateral hemisphere were not reported for mtDNA copynumber (Fig2), TFAM and NDUSF1 protein quantification (Fig2), or BODIPY (Fig3).

The omission of the contralateral western blot and mtDNA data occurred because these data did not reveal significant differences between the treatment groups. However, in response to this request, we have included these data in Supplementary Figure 1. Regarding the BODIPY stain, no staining was observable on the contralateral side, rendering quantification unfeasible. We have addressed this issue in the article in section 2.4 on page 5.

3) In Fig2b, throughout Fig3, in Fig5c, and in Fig6a it is recommended that infarcted be changed to ipsi or ipsilateral for consistency. Also, in some cases IL/CL is used while in other Ispi/Contra is used. This should be made uniform throughout the manuscript and figures.

The labeling of the infarcted side has been revised for consistency, utilizing "Ipsi" or "Contra" consistently throughout. However, "IL" and "CL" have been employed in instances where space constraints prevented the use of "Contra" without reducing font size.

4) The original blots that were provided are not labelled and are not referenced in the text. It is recommended that these blots be included in the supplement and referenced in the text.

The original blots have now been labeled and turned into Supplementary Figure 2. They are now referred to in the text (Section 2.2 on page 3).

Reviewer 2 Report

Comments and Suggestions for Authors

This paper is well written and organized.

The role of neuroinflammation in stroke pathogenesis is widely showed in the recently scientific literature and thos topic have an important scientific sound.

The researchers showed the crucial role of formoterol in mitochondrial activation. This way is important for neuroinflammation shutdown.

I read the paper with interest. The scientific method applied is correct and rigorous.

The figures are clear and original. 

In the future probably the immunomodulatory drugs will be the main post-stroke treatment

Author Response

The authors sincerely thank the reviewer for the encouraging comments.